

# A new hybrid method combining search and direct based construction ideas to generate all 4 × 4 involutory maximum distance separable (MDS) matrices over binary field extensions

Gökhan Tuncay[1], Fatma Büyüksaraçoğlu Sakallı[2], Meltem Kurt Pehlivanoğlu[3], Gülsüm Gözde Yılmazgüç[4], Sedat Akleylek[5,6] and Muharrem Tolga Sakallı[2]

[1] Edirne Vocational College of Technical Sciences, Trakya University, Edirne, Turkey
[2] Department of Computer Engineering, Trakya University, Edirne, Turkey
[3] Department of Computer Engineering, Kocaeli University, Kocaeli, İzmit, Turkey
[4] Ipsala Vocational College, Trakya University, Edirne, Turkey
[5] Cyber Security and Information Technologies Research and Development Center and Department of Computer Engineering, Ondokuz Mayis University, Samsun, Turkey
[6] Institute of Computer Science, University of Tartu, Tartu, Estonia

Corresponding author
Meltem Kurt Pehlivanoğlu,
m.k.kocaeliuni@gmail.com

## ABSTRACT

This article presents a new hybrid method (combining search based methods and direct construction methods) to generate all 4 × 4 involutory maximum distance separable (MDS) matrices over $F_{2^m}$. The proposed method reduces the search space complexity at the level of $\sqrt{n}$, where $n$ represents the number of all 4 × 4 invertible matrices over $F_{2^m}$ to be searched for. Hence, this enables us to generate all 4 × 4 involutory MDS matrices over $F_{2^3}$ and $F_{2^4}$. After applying global optimization technique that supports higher Exclusive-OR (XOR) gates (e.g., XOR3, XOR4) to the generated matrices, to the best of our knowledge, we generate the lightest involutory/non-involutory MDS matrices known over $F_{2^3}$, $F_{2^4}$ and $F_{2^8}$ in terms of XOR count. In this context, we present new 4 × 4 involutory MDS matrices over $F_{2^3}$, $F_{2^4}$ and $F_{2^8}$, which can be implemented by 13 XOR operations with depth 5, 25 XOR operations with depth 5 and 42 XOR operations with depth 4, respectively. Finally, we denote a new property of Hadamard matrix, i.e., (involutory and MDS) Hadamard matrix form is, in fact, a representative matrix form that can be used to generate a small subset of all $2^k \times 2^k$ involutory MDS matrices, where k > 1. For k = 1, Hadamard matrix form can be used to generate all involutory MDS matrices.

## INTRODUCTION

Two important properties used in the design of block ciphers and defined by *Shannon (1949)* are confusion and diffusion. These properties are respectively satisfied by substitution boxes (or shortly S-boxes) and linear transformations in a round function of a block cipher. Maximum distance separable (MDS) matrices are derived from MDS codes

and provide the maximum diffusion. They are used as the core component of diffusion layers/linear transformations in the design of cryptographic primitives like block ciphers and hash functions. MDS matrices have the maximum branch number, which is an important cryptographic criterion used for defining diffusion rate. Branch number also helps to measure security against some well-known attacks like differential (*Biham & Shamir, 1991*) and linear cryptanalysis (*Matsui, 1994*). Block ciphers (or a cryptographic primitive) generally use MDS matrices over $\mathbf{F}_{2^3}$, $\mathbf{F}_{2^4}$ and $\mathbf{F}_{2^8}$ in their diffusion layers according to the design strategies and considering implementation issues of a block cipher. Also, using involutory MDS matrices in the design of a diffusion layer of a block cipher has the advantage of reusing the same circuit in the decryption process and helps to implement a block cipher at close encryption and decryption costs. In this context, we present our experimental results by considering the finite fields $\mathbf{F}_{2^3}$, $\mathbf{F}_{2^4}$ and $\mathbf{F}_{2^8}$. Note that our construction technique can easily be used for designing new involutory MDS matrices over finite field $\mathbf{F}_{2^m}$ especially for $m \leq 8$.

Generally, the construction techniques of MDS matrices can be categorized into three groups: direct construction methods, search based methods, and hybrid methods (combining search based methods and direct construction methods). The direct construction methods include the methods such as Cauchy matrices (*Youssef, Mister & Tavares, 1997*; *Cui, Jin & Kong, 2015*), Vandermonde matrices (*Sajadieh et al., 2012b*) and Companion matrices (*Gupta & Ray, 2013*). It should also be noted that Cauchy and Vandermonde matrices are generally not efficient for low-cost implementations (*Gupta et al., 2019*). Recently, unlike the other direct construction methods, a new direct construction method (or a new matrix form) to generate all $3 \times 3$ involutory MDS matrices has been given in *Güzel et al. (2019)*. On the other hand, search based methods to find MDS matrices are based on using hybrid structures (*Sim et al., 2015*), recursive structures (*Sajadieh et al., 2012a*, *2012b*), and searching some special matrix forms like circulant and Hadamard matrix forms, which have some advantages in the implementation phase and have a higher probability of finding MDS matrix when compared to a randomized square matrix. Moreover, the other special matrix forms used to find (lightweight) MDS matrices are as follows: circulant-like (*Gupta & Ray, 2015*), Toeplitz, Toeplitz-like, and Hankel matrices (*Gupta et al., 2019*). The search based construction methods are useful for finding MDS matrices with small orders. But, finding MDS matrices with higher orders is exactly an NP-complete problem (computation cost for checking a matrix to be MDS is still too expensive) (*Sim et al., 2015*). To handle this problem, the Generalized Hadamard (shortly GHadamard) matrix form, a hybrid construction method, was proposed in *Pehlivanoğlu et al. (2018)*. Overall, in *Sakallı et al. (2020)*, the authors proposed a complementary method for the current construction methods in the literature, which generates isomorphic $k \times k$ MDS matrices (new MDS matrices from the implementation point of view) from any existing $k \times k$ MDS matrix (due to its ground field structure). All these methods can be evaluated within the local optimization category that focuses on the coefficients of a given matrix. In recent years, global optimization techniques have been proposed to construct smaller diffusion layer circuits for involutory/non-involutory MDS matrices. This challenging problem is also

known as the problem of finding the shortest linear straight-line program (SLP) and optimizing circuits globally. In this context, two main global optimization techniques can be given as cancellation-free programs (*e.g.*, Paar's algorithms (*Paar, 1997*) and heuristic techniques (*e.g.*, Boyar-Peralta (BP) algorithm (*Boyar & Peralta, 2010*; *Boyar, Matthews & Peralta, 2012*) RNBP, A1, A2 (*Tan & Peyrin, 2019*). All these heuristics support 2-input XOR gates (XOR2) only, but in *Baksi et al. (2021)* the authors by inspiring the idea given in *Banik, Funabiki & Isobe (2019)* proposed a new version of the original BP heuristic, called BDKCI, that supports higher input XOR gates such as 3-input XOR (XOR3) and 4-input XOR (XOR4) gates. Utilizing higher input XOR gates can result in a lower cost in specific ASIC libraries, so in this article, we use BDKCI heuristic for finding the efficient implementation of a given matrix. Moreover, we consider not only gate count (GC) and circuit depth but also gate equivalent (GE) metric for lightweight implementations. BDKCI heuristic directly calculates the total GEs for each ASIC library (STM 90 nm (ASIC1), STM 65nm (ASIC2), TSMC 65 nm (ASIC3), and STM 130 nm (ASIC4)), more technical details can be found in *Baksi et al. (2021)*.

To the best of our knowledge, there is no known method in the literature to represent/generate all involutory MDS matrices over $\mathbf{F}_{2^m}$. In this study, we present a new hybrid method to generate all $4 \times 4$ involutory MDS matrices over $\mathbf{F}_{2^m}$. We consider the problem of finding lightweight involutory/non-involutory MDS matrices with low implementation costs. In this context, we generate all $4 \times 4$ involutory MDS matrices over $\mathbf{F}_{2^3}$ and $\mathbf{F}_{2^4}$ and evaluate these matrices up to a threshold value with respect to naive XOR count (d-XOR) (*Khoo et al., 2014*; *Jean et al., 2017*), and then we look for the lightest hardware circuits known for $4 \times 4$ involutory MDS matrices in terms of XOR counts and circuit depths by using BDKCI algorithm. Note that when obtaining the lightest $4 \times 4$ non-involutory MDS matrix over $\mathbf{F}_{2^4}$, we take the benefit of the elements of the lightest $4 \times 4$ involutory MDS matrix over $\mathbf{F}_{2^4}$ generated.

In this article, we propose a new hybrid method to generate all $4 \times 4$ involutory MDS matrices over $\mathbf{F}_{2^m}$. The proposed hybrid method consists of two parts: the first part includes searching for all $4 \times 4$ representative involutory MDS matrices and the second part includes generating all $4 \times 4$ involutory MDS matrices using all $4 \times 4$ representative involutory MDS matrices found by search in the first part of the proposed method. Hence, we present the number of all $4 \times 4$ involutory MDS matrices over $\mathbf{F}_{2^3}$ and $\mathbf{F}_{2^4}$. In addition, we give the lightest known involutory/non-involutory MDS matrices over $\mathbf{F}_{2^3}$, $\mathbf{F}_{2^4}$ and $\mathbf{F}_{2^8}$ obtained by using the global optimization algorithm BDKCI, which is an improved version of BP algorithm.

## Motivation and our contribution

The demand for low-cost security design targeting resource-constrained devices has triggered the exploration of lightweight diffusion layers (diffusion layers with low implementation costs). In particular, the circuit area is a crucial criterion for lightweight cryptography in terms of hardware. This means that reducing the cost of hardware implementation is to minimize the number of expensive logical gates (especially XOR operations) and the depth of the circuit (for low latency), which is the number of gates on

the longest circuit path. However, it is not an easy task to construct lightweight MDS matrices, especially involutory ones. Involutory MDS matrices use the same circuit and have the same implementation costs in the encryption and decryption phases. Therefore, the designers have considered the problem of building optimal or close to optimal linear circuits for involutory/non-involutory MDS matrices.

Previous studies mainly focused on the problem of constructing a lightweight MDS matrix from two perspectives: building a locally optimized implementation of an MDS matrix consisting of the coefficients that are easy to evaluate or finding the globally optimized implementation of a given MDS matrix. However, while searching lightweight MDS matrices, most studies use some heuristic searching methods that cannot find all MDS matrices especially involutory ones in a specific finite field (except for the matrix form given in *Güzel et al. (2019)* used for generating all $3 \times 3$ involutory MDS matrices). In this article, for the first time in the literature, we focus on a hybrid method generating all $4 \times 4$ involutory MDS matrices over $\mathbf{F}_{2^m}$ with the lightest known implementation costs because many block ciphers use $4 \times 4$ and $8 \times 8$ (in the form of $2^k \times 2^k$) MDS matrices. For example, the Advanced Encryption Standard (AES) (*Daemen & Rijmen, 2002*) uses a $4 \times 4$ MDS matrix as the main part of its diffusion layer. Moreover, (lightweight) block ciphers to be designed in the future may likely use $4 \times 4$ involutory/non-involutory MDS matrices over $\mathbf{F}_{2^m}$ with nice hardware implementation costs. In this article, we combine local optimization (new hybrid construction method) and global optimization (BDKCI algorithm) techniques. The main contributions of this article can be given as follows:

- A new hybrid construction method to generate all $4 \times 4$ involutory MDS matrices over $\mathbf{F}_{2^m}$ is proposed. The proposed method reduces the search space complexity (the search space defining the total number of all invertible $4 \times 4$ matrices over $\mathbf{F}_{2^m}$) at the level of $\sqrt{n}$, where $n$ represents the number of all $4 \times 4$ invertible matrices to be searched for.
- It is shown that the number of all $4 \times 4$ involutory MDS matrices over $\mathbf{F}_{2^m}$ can be calculated by the formula $\#RIM \times (2^m - 1)^3$, where $\#RIM$ represents the number of all $4 \times 4$ representative involutory MDS matrices. Hence, there are, respectively, 16,464 ($= 48 \times (2^3 - 1)^3$) and 242,514,000 ($= 71,856 \times (2^4 - 1)^3$) $4 \times 4$ involutory and MDS matrices over $\mathbf{F}_{2^3}$ and $\mathbf{F}_{2^4}$.
- The new lightest involutory/non-involutory MDS matrices over $\mathbf{F}_{2^3}$, $\mathbf{F}_{2^4}$ and $\mathbf{F}_{2^8}$ achieved by the proposed hybrid construction method are presented:
  - A new $4 \times 4$ involutory MDS matrix over $\mathbf{F}_{2^4}$ which can be implemented by only 25 XOR operations with depth 5 is presented, whereas the previously known lightest one (*Sarkar & Syed, 2016*) requires 26 XOR operations to be implemented with the same depth using XOR2 and XOR3 gates (please see the matrix $M_4$).
  - A new $4 \times 4$ involutory MDS matrix over $\mathbf{F}_{2^3}$ is presented, which can be implemented by only 13 XOR operations with depth 5 using XOR3 and XOR4 gates (please see the matrix $M_3$).
  - A new $4 \times 4$ non-involutory MDS matrix over $\mathbf{F}_{2^4}$ which can be implemented by only 19 XOR operations with depth 3 is presented, whereas previously known lightest one

(*Sarkar & Syed, 2016*) costs 20 XOR operations the same depth using XOR2, XOR3 and XOR4 gates (please see the matrix $M_5$).

– A new $4 \times 4$ involutory MDS matrix over $\mathbf{F}_{2^8}$ can be implemented by only 52 XOR operations with depth 5 using XOR2 and XOR3 gates. Moreover, the same matrix requires only 42 XOR operations with depth 4 using XOR2, XOR3, and XOR4 gates (please see the matrix $M_6$). While compare to the previous best-known implementations of linear layers of some block ciphers, it improves the results in GCs and GEs.

These findings clearly establish that the circuit implementations of matrices $M_3$, $M_4$, $M_5$ and $M_6$ over the specified fields are superior to those given in previous studies. This aspect also underscores one of the key novelties in our article.

- A new property of Hadamard matrix is denoted, *i.e.*, (involutory and MDS) Hadamard matrix form is, in fact, a representative matrix form that can be used to generate a small amount of all $2^k \times 2^k$ involutory MDS matrices, where $k > 1$. For $k = 1$, Hadamard matrix form can be used to generate all involutory MDS matrices. For $4 \times 4$ Hadamard matrices, we present the matrix form $R_1$ (which has the generic properties of Hadamard matrix, *i.e.*, XOR sum of the elements in any row or column of a Hadamard matrix is equal to 1 and XOR sum of the elements in the main diagonal is equal to 0) given in "Proposed Method" section for finding all $4 \times 4$ representative involutory MDS matrices by search. The matrix form $R_1$ can also be adaptable to $2^k \times 2^k$ Hadamard matrices for $k > 2$. Hence, one can generate new representative involutory MDS matrices over any finite field by using the matrix form $R_1$ or adapted versions of $R_1$ for $2^k \times 2^k$ Hadamard matrices for $k > 2$, which may help us find new involutory MDS matrices (also, with the help of $b_i$ parameters given in Theorem 1) with better implementation properties.
- All the optimization results of matrices are available at https://github.com/mkurtpehlivanoglu/Hybrid_Method.

### Organization

This article is organized as follows: We give some notations, properties of MDS matrices, and two metrics used for identifying lightweightness of an MDS matrix in the "Preliminaries" section. In the "Proposed Method" section, we propose a new hybrid method to generate all $4 \times 4$ matrices over $\mathbf{F}_{2^m}$. Experimental results on the number of all $4 \times 4$ involutory MDS matrices and some examples for $4 \times 4$ involutory MDS matrices over $\mathbf{F}_{2^3}$, $\mathbf{F}_{2^4}$ and $\mathbf{F}_{2^8}$ with the lowest XOR counts are presented in "Experimental Results" section. Finally, we conclude the article in the "Conclusion" section.

## PRELIMINARIES

In this section, we describe the mathematical background needed throughout the article. In this context, some notations and definitions are presented.

The finite field $\mathbf{F}_{2^m}$ consisting of $2^m$ elements is defined by an irreducible polynomial $p(x)$ of degree $m$ over $\mathbf{F}_2$ and is denoted by $\mathbf{F}_2[x]/(p(x))$. The elements of the finite field $\mathbf{F}_{2^m}$ can be represented as polynomials over $\mathbf{F}_2[x]/p(x)$, i.e., $\sum_{i=0}^{m-1} a_i\alpha^i$, where $a_i \in \mathbf{F}_2$ and $\alpha$ is a root (and a primitive element) of $\mathbf{F}_{2^m}$. For simplicity, we denote $\mathbf{F}_{2^m}$ defined by irreducible polynomial $p(x)$ as $\mathbf{F}_{2^m}/p(x)$ (the finite field with $2^m$ elements) and use hexadecimal notation to represent the elements of $\mathbf{F}_{2^m}$ and the irreducible polynomial $p(x)$ used for defining $\mathbf{F}_{2^m}$. As an example, the four-bit string 1110 which can also be represented by `0xe` in hexadecimal notation corresponds to the polynomial $\alpha^3 + \alpha^2 + \alpha$ in $\mathbf{F}_{2^4}$. In the same manner, `0x13` used when denoting $\mathbf{F}_{2^4}/\text{0x13}$ stands for the irreducible polynomial $p(x) = x^4 + x + 1$.

If an $[n, k, d]$ code $C$ reaches the Singleton bound $d = n - k + 1$, then $C$ is called an MDS code (where $d$, $n$, and $k$ represent the length and the number of rows of the generating matrix of the code $C$, respectively). Moreover, generator matrices that produce MDS codes are called MDS matrices. MDS matrices derived from MDS codes provide the maximum diffusion in a block cipher and have the maximum differential and linear branch number ($k + 1$ for $k \times k$ for MDS matrices), which are two important cryptographic criteria for linear transformations. The followings are the properties of an MDS matrix:

- Let $\mathbf{M}$ be a $k \times k$ square matrix. $\mathbf{M}$ is an MDS matrix, if and only if every square sub-matrix of $\mathbf{M}$ is non-singular.
- If $\mathbf{M}$ is a $k \times k$ MDS matrix, the transpose matrix of $\mathbf{M}$ ($\mathbf{M}^T$) is also an MDS matrix.
- Let $c \in \mathbf{F}_{2^m}$ be a non-zero constant and let $\mathbf{M}$ be a $k \times k$ MDS matrix, then the multiplication of a row (or column) of $\mathbf{M}$ by $c$ does not affect the MDS property of $\mathbf{M}$.

If a square matrix $\mathbf{A}$ is its own inverse (i.e., $\mathbf{A} = \mathbf{A}^{-1}$), then the matrix $\mathbf{A}$ is called an involutory matrix. Equivalently, the matrix $\mathbf{A}$ is an involution if and only if $\mathbf{A}^2 = \mathbf{I}$, where $\mathbf{I}$ is the identity matrix.

**Definition 1.** *A $k \times k$ Finite Field Hadamard matrix (simply Hadamard matrix) $\mathbf{H}$ over $\mathbf{F}_{2^m}$ with $k = 2^t$ for $t > 0$ can be expressed as follows:*

$$\mathbf{H} = had(\mathbf{A}_0, \mathbf{A}_1) = \begin{bmatrix} \mathbf{A}_0 & \mathbf{A}_1 \\ \mathbf{A}_1 & \mathbf{A}_0 \end{bmatrix} \tag{1}$$

*where sub-matrices $\mathbf{A}_0$ and $\mathbf{A}_1$ are also $2^{t-1} \times 2^{t-1}$ Hadamard matrices.*

When denoting a $k \times k$ Hadamard matrix, we use the notation $had(a_0, a_1,\ldots, a_{k-1})$, where $a_i \in \mathbf{F}_{2^m}$ for $0 \leq i \leq k - 1$. In this respect, a $4 \times 4$ Hadamard matrix $\mathbf{H}$ can be given as follows:

$$\mathbf{H} = had(a_0, a_1, a_2, a_3) = \begin{bmatrix} a_0 & a_1 & a_2 & a_3 \\ a_1 & a_0 & a_3 & a_2 \\ a_2 & a_3 & a_0 & a_1 \\ a_3 & a_2 & a_1 & a_0 \end{bmatrix} \tag{2}$$

Some important properties of a $k \times k$ Hadamard matrix $\mathbf{H}$ over $\mathbf{F}_{2^m}$ are as follows (*Pehlivanoğlu et al., 2018*):

- $H_{i,j} = a_{i \oplus j}$, where $a_i$ parameters are the entries of the first row of a $k \times k$ Hadamard matrix $H$.
- $H$ is a bi-symmetric matrix, namely, $H = H^T$ and $HJ = JH$ where $J$ is a $k \times k$ exchange matrix ($J_{i,k-i+1} = 1$ and other elements of $J$ are 0),
- $H^2 = c^2 \times I$, where $c = \oplus_{i=0}^{k-1} a_i$ and $I$ is the $k \times k$ identity matrix.

If XOR sum of the first row elements of a $k \times k$ Hadamard matrix $H$ over $\mathbf{F}_{2^m}$ is equal to 1, then $H$ is involutory matrix, *i.e.*, $H^2 = I$ and $H = H^{-1}$, where $I$ is identity matrix and $H^{-1}$ is the inverse of $H$. On the other hand, GHadamard matrix form presented in *Pehlivanoğlu et al. (2018)* satisfies the last property $\left(i.e., H^2 = c^2 \times I, \text{ where } c = \oplus_{i=0}^{k-1} a_i\right)$ given above for a $k \times k$ Hadamard matrix and preserves involutory and MDS properties of a given $k \times k$ involutory and MDS Hadamard matrix. The idea in preserving MDS property of GHadamard matrix form is based on the last property for defining an MDS matrix. A $k \times k$ GHadamard matrix $GH$ is generated by using the combination of non-zero $k - 1$ more $b_i$ parameters and their inverses with a $k \times k$ Hadamard matrix $H$ over $\mathbf{F}_{2^m}$. In this context, a $4 \times 4$ GHadamard matrix $GH$ can be denoted as follows:

$$GH = Ghad\left(a_0, a_1; b_1, a_2; b_2, a_3; b_3\right) = \begin{bmatrix} a_0 & a_1 b_1 & a_2 b_2 & a_3 b_3 \\ a_1 b_1^{-1} & a_0 & a_3 b_1^{-1} b_2 & a_2 b_1^{-1} b_3 \\ a_2 b_2^{-1} & a_3 b_2^{-1} b_1 & a_0 & a_1 b_2^{-1} b_3 \\ a_3 b_3^{-1} & a_2 b_3^{-1} b_1 & a_1 b_3^{-1} b_2 & a_0 \end{bmatrix} \tag{3}$$

We estimate the hardware cost of an (involutory) MDS matrix (*i.e.*, linear transformation) with the number of XOR operations required in hardware implementation which can be described as $x_i \leftarrow x_{a_i} \oplus x_{b_i}$ with $a_i, b_i < i$, where $x_{a_i}$ and $x_{b_i}$ are inputs and some subset of $x_i$s are outputs. In the literature, there are two important approximations of the implementation cost in terms of XOR count: direct XOR (d-XOR) count (*Khoo et al., 2014*) and sequential XOR (s-XOR) count (*Beierle, Kranz & Leander, 2016*). While d-XOR count is defined as the Hamming weight (the number of 1 bits) of the corresponding $mk \times mk$ binary matrix (transformed from $k \times k$ matrix over $\mathbf{F}_{2^m}$) minus $mk$, s-XOR count is defined as the minimum number of XOR operations needed to implement the $mk \times mk$ binary matrix with in-place operations and without extra intermediate computations. Although s-XOR count seems like a better approximation, it causes a high computational cost for optimizing full MDS matrices (*Duval & Leurent, 2018*).

Local and global optimization techniques are used to find optimized implementations of MDS matrices in terms of the required number of XOR operations. The main difference between these two techniques is that the global optimization technique focuses on the optimization of a linear Boolean function of a whole matrix circuit while the local optimization technique focuses on the evaluation of diffusion matrix coefficients. Local optimization techniques do not guarantee finding efficient circuit implementation of an MDS matrix and therefore, more recently, global optimization techniques have been addressed to find well optimized circuits. The idea is based on the reduction of XOR count

which is extracted by the naive implementation of an MDS matrix that contains a lot of repeated calculations.

In this article, while performing global optimization of an (involutory) MDS matrix, the circuit constructed by only XOR gates is handled as a linear Boolean function consisting of $n$ input signals $\{x_0, x_1, \ldots, x_n\}$ and $m$ output signals $\{y_0, y_1, \ldots, y_m\}$ (also called target signals). We also use BDKCI algorithm to build a globally optimized implementation of a $4 \times 4$ (involutory) MDS matrix generated by using the proposed new hybrid method. The aim of BDKCI algorithm is to find efficient circuits with intermediate variables $t_i$s (calculated once) for other computation sequences. These circuits can reuse intermediate variables $t_i$s that lead to reducing the number of gates required. More details about BDKCI heuristic can be found in *Baksi et al. (2021)*.

## PROPOSED METHOD

In this section, we present a new method to generate all $4 \times 4$ involutory MDS matrices over $\mathbf{F}_{2^m}$. The proposed method is a hybrid construction method and is based on the combination of search based and direct based construction methods. The main idea of the proposed method is first to generate all $4 \times 4$ representative involutory MDS matrices by search, and then to obtain all $4 \times 4$ involutory MDS matrices by applying three more non-zero parameters and their inverses to these representative matrices. When generating representative involutory MDS matrices, we take the benefit of generic properties of a Hadamard matrix satisfying the involutory property, *i.e.*, XOR sum of the elements in any row or column of a Hadamard matrix is equal to 1 and XOR sum of the elements in the main diagonal is equal to 0. Note that these properties also force XOR sum of the elements in the anti-diagonal (counter diagonal) to be equal to 0. Then, we can easily define the matrix form $R_1$ in order to search for all representative involutory MDS matrices over $\mathbf{F}_{2^m}$ as follows:

$$R_1 = \begin{bmatrix} r_{11} & r_{12} & r_{13} & r_{11} + r_{12} + r_{13} + 1 \\ r_{21} & r_{22} & r_{12} + r_{13} + r_{21} + r_{31} + r_{32} & r_{12} + r_{13} + r_{22} + r_{31} + r_{32} + 1 \\ r_{31} & r_{32} & r_{33} & r_{31} + r_{32} + r_{33} + 1 \\ r_{11} + r_{21} + r_{31} + 1 & r_{12} + r_{22} + r_{32} + 1 & r_{12} + r_{21} + r_{31} + r_{32} + r_{33} + 1 & r_{11} + r_{22} + r_{33} \end{bmatrix}.$$

The matrix form $R_1$ above for finding representatives involutory MDS matrix is defined by eight parameters $(r_{11}, r_{12}, r_{13}, r_{21}, r_{22}, r_{31}, r_{32}, r_{33})$ over $\mathbf{F}_{2^m} - \{0\}$. Note that when defining the matrix form $R_1$, the finite field element 0 is not considered because of the fact that all entries of an MDS matrix should be non-zero by the first property used for defining an MDS matrix. Hence, the search space for finding representative involutory MDS matrices over $\mathbf{F}_{2^m}$ is approximately obtained as $(2^m - 1)^8$ (by omitting non-invertible or singular matrices). For example, for finding all $4 \times 4$ representative involutory MDS matrices over $\mathbf{F}_{2^4}$, the search space is approximately $(2^4 - 1)^8 \approx 2^{31.25}$. Before proceeding with the details on the matrix form $R_1$, we introduce the matrix form $A_{2 \times 2}$ used for generating all $2 \times 2$ involutory MDS matrices over $\mathbf{F}_{2^m}$ in Remark 2 and Lemma 3 from which the main idea of the article comes.

**Remark 2.** *Consider $2^k \times 2^k$ involutory MDS matrices, where $k$ is a positive integer. For $k = 1$, we show in Lemma 3 that one can directly generate all $2 \times 2$ involutory MDS matrices by the following matrix form:*

$$A_{2 \times 2} = \begin{bmatrix} r_{11} & (r_{11} + 1)b_1 \\ (r_{11} + 1)b_1^{-1} & r_{11} \end{bmatrix}$$

*where $r_{11} \in \mathbf{F}_{2^m} - \{0, 1\}$ and $b_1 \in \mathbf{F}_{2^m} - \{0\}$. Note that the following involutory MDS Hadamard matrix form $RIM_{2 \times 2}$:*

$$RIM_{2 \times 2} = \begin{bmatrix} r_{11} & (r_{11} + 1) \\ (r_{11} + 1) & r_{11} \end{bmatrix}$$

*is also representative involutory MDS matrix form for $2 \times 2$ matrices over $\mathbf{F}_{2^m}$ with the restriction $r_{11} \in \mathbf{F}_{2^m} - \{0, 1\}$. Hence, the representative involutory MDS matrix form $RIM_{2 \times 2}$ can be used to generate all involutory MDS matrices over $\mathbf{F}_{2^m}$.*

**Lemma 3.** *Let $A = [r_{ij}] = \begin{bmatrix} r_{11} & r_{12} \\ r_{21} & r_{22} \end{bmatrix}$ be a $2 \times 2$ matrix over $\mathbf{F}_{2^m}$. If the matrix $A$ is involutory and MDS, then all $2 \times 2$ involutory MDS matrices can be generated by the following matrix form:*

$$A_{2 \times 2}(r_{11}, b_1) = \begin{bmatrix} r_{11} & (r_{11} + 1)b_1 \\ (r_{11} + 1)b_1^{-1} & r_{11} \end{bmatrix}$$

*where $r_{22} = r_{11}$, $r_{12} = (r_{11} + 1)b_1$ and $r_{21} = (r_{11} + 1)b_1^{-1}$ for $b_1 \in \mathbf{F}_{2^m} - \{0\}$ and $r_{11} \in \mathbf{F}_{2^m} - \{0, 1\}$.*

*Proof.* Let $A = \begin{bmatrix} r_{11} & r_{12} \\ r_{21} & r_{22} \end{bmatrix}$ be a $2 \times 2$ involutory matrix with the restriction $r_{11} \neq 0$. Let $c_{ij}$ denote the elements of $A^2$ for $i, j \in \{1, 2\}$, i.e., $c_{ij} = \sum_{k=1}^{2} r_{ik} r_{kj}$. Since $A^2 = I$, where $I$ is the $2 \times 2$ identity matrix, we obtain the following equations (by considering if $i = j$ then $c_{ij} = 1$ and if $i \neq j$ then $c_{ij} = 0$):

$$r_{11}^2 + r_{12}r_{21} = 1 \tag{4}$$
$$r_{11}r_{12} + r_{12}r_{22} = 0 \tag{5}$$
$$r_{21}r_{11} + r_{22}r_{21} = 0 \tag{6}$$
$$r_{21}r_{12} + r_{22}^2 = 1 \tag{7}$$

After adding the Eqs. (4) and (7) given above, we obtain the equation $r_{11}^2 = r_{22}^2$. The equation $r_{11}^2 = r_{22}^2$ can be rewritten as $(r_{11} + r_{22})^2 = 0$, and therefore we obtain $r_{11} = r_{22}$ because the operations are performed in the finite field $\mathbf{F}_{2^m}$. On the other hand, we have $r_{12}r_{21} = r_{11}^2 + 1 = (r_{11} + 1)^2$ from the Eq. (4). Then, $r_{12}$ and $r_{21}$ depending on $r_{11}$ and a new parameter $b_1$ are respectively obtained as $r_{12} = (1 + r_{11})b_1$ and $r_{21} = (1 + r_{11})b_1^{-1}$ with the restrictions $b_1 \in \mathbf{F}_{2^m} - \{0\}$ and $r_{11} \in \mathbf{F}_{2^m} - \{0, 1\}$ so that the matrix form can also satisfy the MDS property (*i.e.*, the determinant of the matrix form $A_{2 \times 2}$ is not equal to 0).

**Remark 4** *Lemma 3 shows that the parameter $b_1$ and its inverse $(b_1^{-1})$ are hidden parameters keeping involutory and MDS property for the matrix form $A_{2 \times 2}$, and also that*

*involutory (MDS) matrices are formed as distinct classes in the search space. Similarly, generating all $4 \times 4$ involutory MDS matrices becomes an easier problem i.e., the problem is first to focus on searching for and finding all representative involutory MDS matrices by using the matrix form $R_1$, and then generate all $4 \times 4$ involutory MDS matrices by using representative involutory MDS matrices obtained and the hidden parameters ($b_1$, $b_2$ and $b_3$ with their inverses $b_1^{-1}$, $b_2^{-1}$ and $b_3^{-1}$) given in Theorem 8.*

In Lemma 6, we give the mathematical background needed to define matrix form $R_1$ used for finding representative involutory MDS matrices, which constitutes the search side of the proposed method and we give the characteristics of $4 \times 4$ representative involutory MDS matrices with Definition 5. Then, in Theorem 8, we present the mathematical background needed for the direct construction side of the proposed method, which is based on three non-zero parameters preserving involutory and MDS property of a $4 \times 4$ representative involutory MDS matrix given.

By Lemma 6, representative involutory MDS matrices are found by searching through all possible candidates using the matrix form $R_1$. This allows us to use 8 parameters ($r_{11}$, $r_{12}$, $r_{13}$, $r_{21}$, $r_{22}$, $r_{31}$, $r_{32}$ and $r_{33}$) (instead of 16 parameters needed for defining all $4 \times 4$ matrices over $\mathbf{F}_{2^m}$) in the search phase of the matrix $R_1$ and thus enables us to reduce the search space complexity from $(2^m - 1)^{16}$ to $(2^m - 1)^{8}$, which is approximately at the level of $\sqrt{n}$, where $n$ represents the number of all invertible $4 \times 4$ matrices over $\mathbf{F}_{2^m}$. In this context, we present the matrix form $R_1$ again below:

$$R_1 = \begin{bmatrix} r_{11} & r_{12} & r_{13} & r_{11} + r_{12} + r_{13} + 1 \\ r_{21} & r_{22} & r_{12} + r_{13} + r_{21} + r_{31} + r_{32} & r_{12} + r_{13} + r_{22} + r_{31} + r_{32} + 1 \\ r_{31} & r_{32} & r_{33} & r_{31} + r_{32} + r_{33} + 1 \\ r_{11} + r_{21} + r_{31} + 1 & r_{12} + r_{22} + r_{32} + 1 & r_{12} + r_{21} + r_{31} + r_{32} + r_{33} + 1 & r_{11} + r_{22} + r_{33} \end{bmatrix}.$$

**Definition 5.** *A representative involutory MDS matrix (or shortly RIM) is a $4 \times 4$ involutory matrix R and satisfies the following conditions: XOR sum of all elements of its main diagonal is equal to 0, XOR sum of any rows (and columns) of R is equal to 1 and the MDS property given in "Preliminaries" section.*

**Lemma 6.** *A representative involutory matrix satisfies the first two conditions given in Definition 5 and these two conditions used to define the matrix form $R_1$ guarantee to find all $4 \times 4$ representative involutory matrices.*

*Proof.* Let $R = [r_{ij}]$ be a $4 \times 4$ involutory and representative matrix such that all $r_{ij} \neq 0$ and let $c_{ij} = \sum_{k=1}^{4} r_{ik} r_{kj}$ denote the elements of $R^2$ for $i, j = 1, 2, 3, 4$. Then $R^2 = I$, where $I$ is the $4 \times 4$ identity matrix. If $i = j$, then $c_{ij} = 1$. Otherwise, $c_{ij} = 0$. Hence, the following equations are satisfied:

$$r_{11}^2 + r_{12}r_{21} + r_{13}r_{31} + r_{14}r_{41} = 1 \tag{8}$$

$$r_{21}r_{12} + r_{22}^2 + r_{23}r_{32} + r_{24}r_{42} = 1 \tag{9}$$

$$r_{31}r_{13} + r_{32}r_{23} + r_{33}^2 + r_{34}r_{43} = 1 \tag{10}$$

$$r_{41}r_{14} + r_{42}r_{24} + r_{43}r_{34} + r_{44}^2 = 1 \tag{11}$$

$$r_{11}r_{12} + r_{12}r_{22} + r_{13}r_{32} + r_{14}r_{42} = 0 \tag{12}$$

$$r_{11}r_{13} + r_{12}r_{23} + r_{13}r_{33} + r_{14}r_{43} = 0 \tag{13}$$

$$r_{11}r_{14} + r_{12}r_{24} + r_{13}r_{34} + r_{14}r_{44} = 0 \tag{14}$$

$$r_{21}r_{11} + r_{22}r_{21} + r_{23}r_{31} + r_{24}r_{41} = 0 \tag{15}$$

$$r_{21}r_{13} + r_{22}r_{23} + r_{23}r_{33} + r_{24}r_{43} = 0 \tag{16}$$

$$r_{21}r_{14} + r_{22}r_{24} + r_{23}r_{34} + r_{24}r_{44} = 0 \tag{17}$$

$$r_{31}r_{11} + r_{32}r_{21} + r_{33}r_{31} + r_{34}r_{41} = 0 \tag{18}$$

$$r_{31}r_{12} + r_{32}r_{22} + r_{33}r_{32} + r_{34}r_{42} = 0 \tag{19}$$

$$r_{31}r_{14} + r_{32}r_{24} + r_{33}r_{34} + r_{34}r_{44} = 0 \tag{20}$$

$$r_{41}r_{11} + r_{42}r_{21} + r_{43}r_{31} + r_{44}r_{41} = 0 \tag{21}$$

$$r_{41}r_{12} + r_{42}r_{22} + r_{43}r_{32} + r_{44}r_{42} = 0 \tag{22}$$

$$r_{41}r_{13} + r_{42}r_{23} + r_{43}r_{33} + r_{44}r_{43} = 0 \tag{23}$$

By adding the Eqs. (8)–(11) side by side, the following equation is obtained:

$$r_{11}{}^2 + r_{22}{}^2 + r_{33}{}^2 + r_{44}{}^2 = 0 \tag{24}$$

Because we study in the finite field $\mathbf{F}_{2^m}$ the Eq. (24) can be rewritten as follows:

$$r_{11} + r_{22} + r_{33} + r_{44} = 0 \tag{25}$$

Hence, we verify the first statement of the lemma. To show the proof of the second statement, we assume that the sum of elements of any row of $R$ is equal to 1. We prefer to study with rows instead of columns of $R$ without breaking the generality. Then, the equations corresponding to our assumption are obtained as follows:

$$r_{11} + r_{12} + r_{13} + r_{14} = 1 \tag{26}$$

$$r_{21} + r_{22} + r_{23} + r_{24} = 1 \tag{27}$$

$$r_{31} + r_{32} + r_{33} + r_{34} = 1 \tag{28}$$

$$r_{41} + r_{42} + r_{43} + r_{44} = 1 \tag{29}$$

By adding the Eqs. (12)–(14) side by side, we get the following equality:

$$r_{11}(r_{12} + r_{13} + r_{14}) + r_{12}(r_{22} + r_{23} + r_{24}) + r_{13}(r_{32} + r_{33} + r_{34}) + r_{14}(r_{42} + r_{43} + r_{44}) = 0 \tag{30}$$

From the assumption Eqs. (26)–(28), it is clear that:

$$r_{11} + 1 = r_{12} + r_{13} + r_{14} \tag{31}$$

$$r_{21} + 1 = r_{22} + r_{23} + r_{24} \tag{32}$$

$$r_{31} + 1 = r_{32} + r_{33} + r_{34} \tag{33}$$

$$r_{41} + 1 = r_{42} + r_{43} + r_{44} \tag{34}$$

By replacing the expressions $r_{12} + r_{13} + r_{14}$, $r_{22} + r_{23} + r_{24}$, $r_{32} + r_{33} + r_{34}$ and $r_{42} + r_{43} + r_{44}$ in the Eq. (30) with the expressions $r_{11} + 1$, $r_{21} + 1$, $r_{31} + 1$ and $r_{41} + 1$, respectively, the Eq. (35) and then the Eq. (36) are obtained:

$$r_{11}(r_{11}+1)+r_{12}(r_{21}+1)+r_{13}(r_{31}+1)+r_{14}(r_{41}+1)=0 \tag{35}$$

$${r_{11}}^2+r_{11}+r_{12}r_{21}+r_{12}+r_{13}r_{31}+r_{13}+r_{14}r_{41}+r_{14}=0 \tag{36}$$

We can rewrite the Eq. (36) as follows:

$${r_{11}}^2+r_{12}r_{21}+r_{12}+r_{13}r_{31}+r_{14}r_{41}=r_{11}+r_{12}+r_{13}+r_{14} \tag{37}$$

Then, by using the assumption (26), we can obtain the Eq. (8) as follows:

$${r_{11}}^2+r_{12}r_{21}+r_{12}+r_{13}r_{31}+r_{14}r_{41}=1 \tag{38}$$

In a similar manner, the Eqs. (9)–(11) can be obtained by using the equations from (15) to (23) and the assumptions (26)–(29).

Similar assumptions can be given when proving the second part of the lemma, related to the sum of the elements of any column, as follows:

$$r_{11}+r_{21}+r_{31}+r_{41}=1 \tag{39}$$

$$r_{12}+r_{22}+r_{32}+r_{42}=1 \tag{40}$$

$$r_{13}+r_{23}+r_{33}+r_{43}=1 \tag{41}$$

$$r_{14}+r_{24}+r_{34}+r_{44}=1 \tag{42}$$

One can easily show that the Eqs. (8)–(11) can again be obtained by using the equations from (12) to (23) and the assumption Eqs. (39)–(42). For instance, the Eq. (8) can be obtained by applying similar operations shown in proving the first part of the lemma and by using the assumption equations from (39) to (42) and the Eqs. (15), (18) and (21). Hence, the second part of the lemma is satisfied.

**Remark 7.** *Lemma 6 does not guarantee that any matrix satisfying assumptions is exactly a representative involutory matrix.*

**Theorem 8.** *Let $RIM = [r_{ij}]$ be a $4 \times 4$ representative involutory MDS matrix, then the matrix form $A = [a_{ij}]$ obtained by the matrix RIM and some parameters ($b_i$ parameters) is also involutory MDS in the following form:*

$$A = [a_{ij}] = \begin{bmatrix} r_{11} & r_{12}b_1 & r_{13}b_2 & r_{14}b_3 \\ r_{21}b_1^{-1} & r_{22} & r_{23}b_1^{-1}b_2 & r_{24}b_1^{-1}b_3 \\ r_{31}b_2^{-1} & r_{32}b_2^{-1}b_1 & r_{33} & r_{34}b_2^{-1}b_3 \\ r_{41}b_3^{-1} & r_{42}b_3^{-1}b_1 & r_{43}b_3^{-1}b_2 & r_{44} \end{bmatrix}$$

*where $b_i$s for $i \in \{1,2,3\}$ are the elements of $\mathbf{F}_{2^m} - \{0\}$.*

*Proof.* Since the representative MDS matrix *RIM* is involutory, it satisfies all equations from (8) to (23) given in Lemma 6. If we square the matrix $A = [a_{ij}]$, we obtain the following expressions corresponding to $c_{ij}$'s, where $c_{ij}$ denote the elements of $A^2$ for $i,j \in \{1,2,3,4\}$:

$$c_{11} = {r_{11}}^2+r_{12}r_{21}+r_{13}r_{31}+r_{14}r_{41} \tag{43}$$

$$c_{22} = r_{21}r_{12}+{r_{22}}^2+r_{23}r_{32}+r_{24}r_{42} \tag{44}$$

$$c_{33} = r_{31}r_{13} + r_{32}r_{23} + r_{33}{}^2 + r_{34}r_{43} \tag{45}$$

$$c_{44} = r_{41}r_{14} + r_{42}r_{24} + r_{43}r_{34} + r_{44}{}^2 \tag{46}$$

$$c_{12} = b_1(r_{11}r_{12} + r_{12}r_{22} + r_{13}r_{32} + r_{14}r_{42}) \tag{47}$$

$$c_{13} = b_2(r_{11}r_{13} + r_{12}r_{23} + r_{13}r_{33} + r_{14}r_{43}) \tag{48}$$

$$c_{14} = b_3(r_{11}r_{14} + r_{12}r_{24} + r_{13}r_{34} + r_{14}r_{44}) \tag{49}$$

$$c_{21} = b_1^{-1}(r_{21}r_{11} + r_{22}r_{21} + r_{23}r_{31} + r_{24}r_{41}) \tag{50}$$

$$c_{23} = b_1^{-1}b_2(r_{21}r_{13} + r_{22}r_{23} + r_{23}r_{33} + r_{24}r_{43}) \tag{51}$$

$$c_{24} = b_1^{-1}b_3(r_{21}r_{14} + r_{22}r_{24} + r_{23}r_{34} + r_{24}r_{44}) \tag{52}$$

$$c_{31} = b_2^{-1}(r_{31}r_{11} + r_{32}r_{21} + r_{33}r_{31} + r_{34}r_{41}) \tag{53}$$

$$c_{32} = b_2^{-1}b_1(r_{31}r_{12} + r_{32}r_{22} + r_{33}r_{32} + r_{34}r_{42}) \tag{54}$$

$$c_{34} = b_2^{-1}b_3(r_{31}r_{14} + r_{32}r_{24} + r_{33}r_{34} + r_{34}r_{44}) \tag{55}$$

$$c_{41} = b_3^{-1}(r_{41}r_{11} + r_{42}r_{21} + r_{43}r_{31} + r_{44}r_{41}) \tag{56}$$

$$c_{42} = b_3^{-1}b_1(r_{41}r_{12} + r_{42}r_{22} + r_{43}r_{32} + r_{44}r_{42}) \tag{57}$$

$$c_{43} = b_3^{-1}b_2(r_{41}r_{13} + r_{42}r_{23} + r_{43}r_{33} + r_{44}r_{43}) \tag{58}$$

The expressions from (43) to (46) are equal to 1 and the expressions from (47) to (58) are equal to 0 because the expressions from (43) to (46) are, respectively, the same with the left side of the equations from (8) to (11) given in Lemma 2, the expressions from (47) to (58) contain the results of multiplying 0 by non-zero finite field element(s) because the same expressions from (12) to (23) in Lemma 6, which are equal to 0, appear respectively in the expressions from (47) to (58). Hence, the involutory property of the matrix *RIM* is preserved. On the other hand, $b_i$ parameters in the matrix form $A$ can be obtained by multiplying the constant elements $c_1, c_2, c_3$ and $c_4 \in \mathbf{F}_{2^m} - \{0\}$ with the first column, the second column, the third column and the fourth column, respectively, and by multiplying the inverses of these elements $c_1^{-1}, c_2^{-1}, c_3^{-1}$ and $c_4^{-1}$ with the first row, the second row, the third row and the fourth row respectively. As given in "Preliminaries" section, the MDS property is invariant under the multiplication of a row/column with a non-zero constant. Hence, the MDS property of the matrix *RIM* is also preserved.

**Remark 9.** *By Theorem 8, one can directly obtain only one representative involutory (MDS) matrix starting from any involutory (MDS) matrix by using $b_i$ parameters and their inverses given in the matrix form A.*

The proposed method can be divided into two stages as follows:

- By Lemma 6, representative involutory MDS matrices are found by searching through all possible candidates using the matrix form $R_1$, which constitutes the search side of the proposed method. This allows us to use 8 parameters ($r_{11}, r_{12}, r_{13}, r_{21}, r_{22}, r_{31}, r_{32}$ and $r_{33}$) in the search phase of the matrix $R_1$. Hence, the search space for finding representative involutory MDS matrices over $\mathbf{F}_{2^m}$ is $(2^m - 1)^8$ by excluding the finite field element 0 in all parameters (the finite field element 0 is ignored in the search space calculation because all entries of an MDS matrix should be non-zero). However, when searching for finding all $4 \times 4$ involutory MDS matrices in a conventional manner, 16 parameters are

needed to define a $4 \times 4$ finite field matrix, which makes the search space of a conventional search $(2^m - 1)^{16}$. Thus, our hybrid method reduces the search space complexity from approximately $(2^m - 1)^{16}$ to $(2^m - 1)^8$, which is approximately at the level of $\sqrt{n}$, where $n$ represents the number of all invertible $4 \times 4$ matrices over $\mathbf{F}_{2^m}$.

- By Theorem 8, all $4 \times 4$ involutory MDS matrices are generated directly by using representative involutory MDS matrices found by search in the previous stage and $b_i$ parameters.

In Examples 10 and 11, we obtain $4 \times 4$ involutory MDS matrices over $\mathbf{F}_{2^4}$ by using the proposed method, which is also given in the literature.

**Example 10.** *Let $\mathbf{F}_{2^4}$ be generated by the primitive element $\alpha$ which is a root of the primitive polynomial $x^4 + x + 1$ (0x13). Consider the $4 \times 4$ $\theta$-circulant involutory MDS matrix $M_1$ recently given in* Cauchois & Loidreau (2019).

$$M_1 = \begin{bmatrix} \alpha & 1 & \alpha^{14} & \alpha^7 \\ \alpha^{14} & \alpha^2 & 1 & \alpha^{13} \\ \alpha^{11} & \alpha^{13} & \alpha^4 & 1 \\ 1 & \alpha^7 & \alpha^{11} & \alpha^8 \end{bmatrix}$$

*over $\mathbf{F}_{2^4}/$0x13. In fact, the involutory MDS matrix $M_1$ belongs to a class of which representative involutory MDS matrix is as follows:*

$$RIM_1 = \begin{bmatrix} \alpha & \alpha^7 & \alpha^5 & \alpha^{11} \\ \alpha^7 & \alpha^2 & \alpha^{14} & \alpha^{10} \\ \alpha^5 & \alpha^{14} & \alpha^4 & \alpha^{13} \\ \alpha^{11} & \alpha^{10} & \alpha^{13} & \alpha^8 \end{bmatrix}$$

*which is symmetric and is also $4 \times 4$ $\theta$-circulant involutory MDS matrix. The matrix $M_1$ can easily be obtained by applying the parameters $b_1 = \alpha^8$, $b_2 = \alpha^9$ and $b_3 = \alpha^{11}$ (and their inverses) to representative involutory MDS matrix $RIM_1$.*

**Example 11.** *Let $\mathbf{F}_{2^4}$ be generated by the primitive element $\alpha$ which is a root of the primitive polynomial $x^4 + x + 1$ (0x13). Consider the $4 \times 4$ involutory MDS matrix $M_2$ given in* Pehlivanoğlu et al. (2018).

$$M_2 = \begin{bmatrix} 1 & 1 & 1 & 1 \\ 1 & \alpha & \alpha^2 & \alpha^5 \\ \alpha^7 & \alpha^{10} & \alpha & \alpha^5 \\ \alpha^9 & \alpha^2 & \alpha^{10} & 1 \end{bmatrix}$$

*over $\mathbf{F}_{2^4}/$0x13. In fact, the involutory MDS matrix $M_2$ belongs to a class of which representative involutory MDS matrix is as follows:*

$$RIM_2 = \begin{bmatrix} 1 & \alpha^6 & \alpha^2 & \alpha^3 \\ \alpha^9 & \alpha & \alpha^{13} & \alpha^2 \\ \alpha^5 & \alpha^{14} & \alpha & \alpha^6 \\ \alpha^6 & \alpha^5 & \alpha^9 & 1 \end{bmatrix}$$

*The involutory MDS matrix $M_2$ can easily be obtained by applying the parameters $b_1 = \alpha^9$, $b_2 = \alpha^{13}$ and $b_3 = \alpha^{12}$ (and their inverses) to representative involutory MDS matrix $RIM_2$.*

## EXPERIMENTAL RESULTS

In this section, we generate all $4 \times 4$ involutory MDS matrices over $\mathbf{F}_{2^3}$ and $\mathbf{F}_{2^4}$ by finding all representative involutory MDS matrices over these finite fields. In order to generate all representative involutory MDS matrices, we use the the matrix form $R_1$ given in the "Proposed Method" section.

The experimental results show that there are 48 and 71,856 representative involutory and MDS matrices over $\mathbf{F}_{2^3}$ and $\mathbf{F}_{2^4}$, respectively. After applying $b_i$ parameters to representative involutory MDS matrices, we generated totally $48 \cdot (2^3 - 1)^3 = 16,464$ and $71,856 \cdot (2^4 - 1)^3 = 242,514,000 \approx 2^{27.85}$ $4 \times 4$ involutory and MDS matrices over $\mathbf{F}_{2^3}$ and $\mathbf{F}_{2^4}$, respectively. Note that one can obtain 24 and 1,512 involutory MDS matrices over $\mathbf{F}_{2^3}$ and $\mathbf{F}_{2^4}$ by searching and using Hadamard matrix form, which are, in fact, a small amount of representative involutory MDS matrices. Then, by using GHadamard matrix form given in *Pehlivanoğlu et al. (2018)*, one can totally generate $24 \cdot (2^3 - 1)^3 = 8,232$ and $1,512 \cdot (2^4 - 1)^3 = 5,103,000 \approx 2^{22.28}$ $4 \times 4$ involutory and MDS matrices over these finite fields.

From the experimental results given above, it is shown that (involutory and MDS) Hadamard matrix form, which is also representative (involutory and MDS) matrix form, can approximately generate 2.1% of all involutory MDS matrices over $\mathbf{F}_{2^4}$. It is likely that this percentage will reduce for involutory MDS matrices over larger finite fields. Since Lemma 6 and Theorem 8 given in "Proposed Method" section can be updated for $2^k \times 2^k$ involutory matrices, it is clear that (involutory and MDS) Hadamard matrix form can be used to generate a small subset of all $2^k \times 2^k$ involutory MDS matrices, where $k > 1$, especially over larger finite fields.

**Remark 12.** *As stated above, there are 71,856 $4 \times 4$ representative involutory MDS matrices over $\mathbf{F}_{2^4}$, 1,512 of which can also be obtained by $4 \times 4$ Hadamard matrix (by search). Then, 70,344 ($= 71,856 - 1,512$) $4 \times 4$ representative involutory MDS matrices cannot be generated by $4 \times 4$ Hadamard matrix form. That means, by using the method given in Sakallı et al. (2020), one can map these representative involutory MDS matrices in order to obtain directly isomorphic counterparts over $\mathbf{F}_{2^8}$. Note that there are 4 isomorphisms from the finite field $\mathbf{F}_{2^4}$ (defined by any irreducible polynomial) to the finite field $\mathbf{F}_{2^8}$ (defined by any irreducible polynomial). Hence, by using the parameters $b_1$, $b_2$, $b_3 \in \mathbf{F}_{2^m} - \{0\}$ and their inverses, one can directly generate 4,665,600,972,000 ($= (255)^3 \cdot 70,344 \cdot 4 \approx 2^{42.08}$) $4 \times 4$ involutory MDS matrices over $\mathbf{F}_{2^8}$ (defined by any irreducible polynomial), which cannot also be generated by GHadamard matrix.*

In Tables 1 and 2, we present the number of occurrences of 1s in all $4 \times 4$ involutory MDS matrices over $\mathbf{F}_{2^3}$ and $\mathbf{F}_{2^4}$, respectively.

**Remark 13.** *In Junod & Vaudenay (2005b), the maximum number of occurrences of 1s in $4 \times 4$ MDS matrices is shown to be 9. In this article, we modify this result by showing that*

**Table 1 The number of occurrences of 1s in all $4 \times 4$ involutory MDS matrices over $F_{2^3}$.**

|  | 0 | 1 | 2 | 3 | 4 | 5 | 6 | 7 | 8 | 9 |
|---|---|---|---|---|---|---|---|---|---|---|
| The number of matrices | 1,368 | 2,424 | 4,608 | 3,600 | 1,944 | 1,296 | 720 | 432 | 0 | 72 |

**Table 2 The number of occurrences of 1s in all $4 \times 4$ involutory MDS matrices over $F_{2^4}$.**

|  | 0 | 1 | 2 | 3 | 4 | 5 | 6 | 7 | 8 | 9 |
|---|---|---|---|---|---|---|---|---|---|---|
| The number of matrices | 73,266,816 | 88,442,736 | 53,722,608 | 20,148,576 | 5,555,760 | 1,146,768 | 206,160 | 21,120 | 3,264 | 192 |

*the maximum number of occurrences of 1s for $4 \times 4$ involutory and MDS matrices is also 9 (see "Appendix" section).*

In this article, we generate the lightest $4 \times 4$ involutory MDS matrices over $F_{2^3}$, $F_{2^4}$ and $F_{2^8}$ in terms of XOR count. To do so, we first generate involutory MDS matrices with low naive XOR counts. In this context, we consider all $4 \times 4$ involutory MDS matrices over $F_{2^3}$ and consider $4 \times 4$ involutory MDS matrices over $F_{2^4}$ with up to a threshold naive XOR count. For $4 \times 4$ involutory MDS matrices over $F_{2^8}$, we consider anti-diagonal symmetric matrices with up to a threshold naive XOR count and choose among them the ones whose results are up to a maximum of 110 XOR operations after optimizing with PAAR 1. Finally, $4 \times 4$ involutory MDS matrices suitable for the given criteria are optimized with BDKCI algorithm.

In Example 14, we present a new $4 \times 4$ involutory MDS matrix over $F_{2^3}$ generated by using the proposed method, which can be implemented by only 13 XOR operations with depth 6.

**Example 14.** *Let $F_{2^3}$ be generated by the primitive element $\alpha$ which is a root of the primitive polynomial $x^3 + x^2 + 1$ (0xd). Consider the $4 \times 4$ involutory MDS matrix $M_3$ as given below:*

$$M_3 = \begin{bmatrix} 1 & 1 & 1 & 1 \\ \alpha^5 & \alpha & \alpha^2 & 1 \\ \alpha^3 & 1 & \alpha^2 & \alpha^4 \\ \alpha^6 & \alpha & 1 & \alpha^4 \end{bmatrix}$$

*over $F_{2^3}/0xd$. In fact, the involutory MDS matrix $M_3$ belongs to a class of which representative involutory MDS matrix is as follows:*

$$RIM_3 = \begin{bmatrix} 1 & \alpha^6 & \alpha^5 & \alpha^3 \\ \alpha^6 & \alpha & \alpha & \alpha^4 \\ \alpha^5 & \alpha & \alpha^2 & \alpha^2 \\ \alpha^3 & \alpha^4 & \alpha^2 & \alpha^4 \end{bmatrix}$$

*The involutory MDS matrix $M_3$ with d-XOR count 56 ($= 20 + 4 \cdot 3 \cdot 3$) can easily be obtained by applying the parameters $b_1 = \alpha$, $b_2 = \alpha^2$ and $b_3 = \alpha^4$ (and their inverses) to representative involutory MDS matrix $RIM_3$. After applying BDKCI heuristic to the matrix*

$M_3$, we find the circuits for the matrix $M_3$ with 13 XORs and depth 5 using XOR3 and XOR4 gates (please see *Table 5* for details).

In Example 15, we present a new $4 \times 4$ involutory MDS matrix over $\mathbf{F}_{2^4}$ by using the proposed method, which is better than the matrices given in the literature. It can be implemented by only 25 XOR operations with depth 5, whereas the previously known lightest one (*Sarkar & Syed, 2016*) requires 26 XOR operations with the same depth using XOR2 and XOR3 gates.

**Example 15.** *Let* $\mathbf{F}_{2^4}$ *be generated by the primitive element* $\alpha$ *which is a root of the primitive polynomial* $x^4 + x + 1$ *(0x13). Consider* $4 \times 4$ *involutory MDS matrix* $M_4$ *as given below:*

$$M_4 = \begin{bmatrix} 1 & \alpha^4 & \alpha^{14} & \alpha \\ 1 & \alpha^{14} & \alpha & \alpha^{14} \\ \alpha^8 & \alpha & \alpha^{14} & \alpha^4 \\ \alpha^2 & \alpha^8 & 1 & 1 \end{bmatrix}$$

*over* $\mathbf{F}_{2^4}/$ *0x13. In fact, the involutory MDS matrix* $M_4$ *belongs to a class of which representative involutory MDS matrix is as follows:*

$$RIM_4 = \begin{bmatrix} 1 & 1 & \alpha^9 & \alpha^7 \\ \alpha^4 & \alpha^{14} & 1 & \alpha^9 \\ \alpha^{13} & \alpha^2 & \alpha^{14} & 1 \\ \alpha^{11} & \alpha^{13} & \alpha^4 & 1 \end{bmatrix}$$

*The involutory MDS matrix* $M_4$ *with d-XOR count 79 ($= 31 + 4 \times 3 \times 4$) can easily be obtained by applying the parameters* $b_1 = \alpha^4$, $b_2 = \alpha^5$ *and* $b_3 = \alpha^9$ *(and their inverses) to representative involutory MDS matrix* $RIM_4$. *After applying BDKCI heuristic to the matrix* $M_4$, *we find the circuits for the matrix* $M_4$ *with 25 XORs and depth 5 using XOR2 and XOR3 gates (please see* Table 6 *for details).*

In Example 16, we present a new $4 \times 4$ non-involutory MDS matrix over $\mathbf{F}_{2^4}$. We have found this matrix by searching through all anti-diagonal symmetric matrices with the same elements of the $4 \times 4$ involutory MDS matrix presented in Example 15. This non-involutory MDS matrix can be implemented by only 19 XOR operations with depth 3.

**Example 16.** *Let* $\mathbf{F}_{2^4}$ *be generated by the primitive element* $\alpha$ *which is a root of the primitive polynomial* $x^4 + x + 1$ *(0x13). Consider the* $4 \times 4$ *anti-diagonal symmetric non-involutory MDS matrix* $M_5$ *as given below:*

$$M_5 = \begin{bmatrix} \alpha^{14} & 1 & \alpha^2 & 1 \\ 1 & \alpha^8 & \alpha & \alpha^2 \\ 1 & \alpha^{14} & \alpha^8 & 1 \\ 1 & 1 & 1 & \alpha^{14} \end{bmatrix}$$

*over* $\mathbf{F}_{2^4}/$ *0x13. The non-involutory MDS matrix* $M_5$ *is with d-XOR count 68 ($= 20 + 4 \times 3 \times 4$). After applying BDKCI heuristic to* $M_5$, *we find the circuits with 19 XORs and depth 3 using XOR2, XOR3 and XOR4 gates.*

**Table 3 Comparison of $4 \times 4$ involutory and non-involutory MDS matrices in view of XOR counts with different depths obtained by using BDKCI heuristic.**

| Over $F_{2^m}$/Poly | Cost | | | | | | | | | Ref. |
|---|---|---|---|---|---|---|---|---|---|---|
| | # XOR2 | # XOR3 | # XOR4 | Depth | GC | ASIC1(GE) | ASIC2(GE) | ASIC3(GE) | ASIC4(GE) | |
| Involutory | | | | | | | | | | |
| $F_{2^4}$/0x13 | 10 | 20 | – | 4 | 30 | 85 | 94.11 | 109 | 126.5 | *Sim et al. (2015)* |
| $F_{2^4}$/0x13 | 4 | 22 | – | 5 | 26 | 79.5 | 89.654 | 102.4 | 115.84 | *Sarkar & Syed (2016)* |
| $F_{2^4}$/0x13 | 5 | 23 | – | 4 | 28 | 84.75 | 95.35 | 109.1 | 123.83 | *Jean et al. (2017)* |
| $F_{2^4}$/0x13 | 4 | 21 | – | 5 | **25** | **76.25** | **85.939** | **98.2** | **111.18** | Exam. 15 |
| $F_{2^3}$/0xd | – | 2 | 11 | 3 | 13 | 61.5 | 67.93 | 77.15 | 75.21 | *Pehlivanoğlu et al. (2023)* |
| $F_{2^3}$/0xd | – | 5 | 8 | 5 | **13** | **56.25** | **62.575** | **71** | **71.22** | Exam. 14 |
| Non-involutory | | | | | | | | | | |
| $F_{2^4}$/0x13 | – | 10 | 12 | 5 | 22 | 92.5 | 103.15 | 117 | 118.48 | *Sim et al. (2015)* |
| $F_{2^4}$/0x13 | – | 6 | 14 | 4 | 20 | 89.5 | 99.29 | 112.7 | 111.82 | *Liu & Sim (2016)* |
| $F_{2^4}$/0x13 | 4 | 1 | 15 | 4 | 20 | 86.25 | 94.139 | 107.95 | 107.83 | *Beierle, Kranz & Leander (2016)* |
| $F_{2^4}$/0x19 | – | 6 | 14 | 4 | 20 | 89.5 | 99.29 | 112.7 | 111.82 | *Beierle, Kranz & Leander (2016)* |
| $F_{2^4}$/0x19 | 2 | 5 | 13 | 3 | 20 | 85.25 | 94.037 | 107.25 | 107.83 | *Sarkar & Syed (2016)* |
| $F_{2^4}$/0x13 | 2 | 5 | 13 | 5 | 20 | 85.25 | 94.037 | 107.25 | 107.83 | *Jean et al. (2017)* |
| $F_{2^4}$/0x13 | 3 | 3 | 13 | 9 | 19 | 80.75 | 88.588 | 101.35 | 101.84 | *Sajadieh & Mousavi (2021)* |
| $F_{2^4}$/0x19 | 1 | 2 | 15 | 10 | 18 | 83.5 | 91.911 | 104.65 | 102.5 | *Sajadieh & Mousavi (2021)* |
| $F_{2^4}$/0x13 | 1 | 8 | 10 | 3 | **19** | **78** | **86.701** | **98.6** | **100.51** | Exam. 16 |

Bold values indicate the best results.

**Table 4 Comparison of best known implementation cost of few $32 \times 32$ matrices (linear layers of block ciphers) in ASIC libraries.**

| Matrix | Cost | | | | | | | |
|---|---|---|---|---|---|---|---|---|
| | # XOR2 | # XOR3 | # XOR4 | GC | ASIC1(GE) | ASIC2(GE) | ASIC3(GE) | ASIC4(GE) |
| AES | 31 | 26 | 4 | 61 | 166.5 *Liu et al. (2022)* | – | – | – |
| | 22 | 21 | 12 | 55 | – | – | – | 243 *Liu et al. (2022)* |
| ANUBIS *Barreto & Rijmen (2000)* | 28 | 26 | 7 | 61 | 175.5 *Liu et al. (2022)* | – | – | – |
| | 21 | 24 | 12 | 57 | – | – | – | 253.6 *Liu et al. (2022)* |
| CLEFIA $M_0$ *Shirai et al. (2007)* | 32 | 32 | 2 | 66 | 178 *Liu et al. (2022)* | – | – | – |
| | 23 | 16 | 18 | 57 | – | – | – | 258.9 *Liu et al. (2022)* |
| CLEFIA $M_1$ *Shirai et al. (2007)* | 35 | 31 | 3 | 69 | 185.7 *Liu et al. (2022)* | – | – | – |
| | 20 | 27 | 13 | 60 | – | – | – | 270.2 *Liu et al. (2022)* |
| FOX MU4 *Junod & Vaudenay (2005a)* | 46 | 43 | – | 89 | 231.7 *Banik, Funabiki & Isobe, 2021* | – | – | – |
| | 32 | 26 | 20 | 78 | – | – | – | 347.5 *Liu et al. (2022)* |
| Exam. 17 | 8 | 44 | – | **52 (depth 5)** | **159** | 179.308 | 204.8 | 231.68 |
| | 3 | 10 | 29 | **42 (depth 4)** | 183.5 | 202.593 | 230.75 | **230.3** |

Bold values indicate the best results.

**Table 5** The global optimization result of $M_3$ with 13 XORs and depth 5, where $x_i$s $[(x_0, x_1, \ldots, x_{11})]$, $y_i$s $[(y_0, y_1, \ldots, y_{11})]$ and $t_j$s represent input bits, output bits, and temporary variables, respectively.

| No. | Operation | No. | Operation |
|---|---|---|---|
| 1 | $y_1 = x_1 + x_4 + x_7 + x_{10}$ | 8 | $y_7 = x_5 + x_{11} + y_{11}$ |
| 2 | $y_0 = x_0 + x_3 + x_6 + x_9$ | 9 | $y_5 = y_2 + y_8 + y_{11}$ |
| 3 | $y_2 = x_2 + x_5 + x_8 + x_{11}$ | 10 | $y_9 = x_8 + x_{11} + y_1 + y_5$ |
| 4 | $t_3 = x_0 + x_1 + x_5 + x_8$ | 11 | $y_6 = x_4 + x_{10} + y_7 + y_{10}$ |
| 5 | $y_8 = x_3 + x_7 + y_0 + t_3$ | 12 | $y_3 = x_0 + x_6 + y_8$ |
| 6 | $y_{10} = x_3 + x_6 + y_2 + y_8$ | 13 | $y_4 = x_1 + x_7 + y_6$ |
| 7 | $y_{11} = x_4 + x_9 + t_3$ | | |

**Table 6** The global optimization result of $M_4$ with 25 XORs and depth 5, where $x_i$s $[(x_0, x_1, \ldots, x_{15})]$, $y_i$s $[(y_0, y_1, \ldots, y_{15})]$ and $t_j$s represent input bits, output bits, and temporary variables, respectively.

| No. | Operation | No. | Operation |
|---|---|---|---|
| 1 | $t_0 = x_7 + x_{15}$ | 14 | $t_{13} = x_1 + t_0 + y_7$ |
| 2 | $y_6 = x_2 + x_9 + t_0$ | 15 | $y_9 = x_2 + x_{13} + t_{13}$ |
| 3 | $t_2 = x_3 + x_4 + x_{10}$ | 16 | $y_1 = x_3 + x_5 + t_{13}$ |
| 4 | $y_7 = x_{12} + t_2$ | 17 | $t_{16} = x_2 + x_4 + x_8$ |
| 5 | $t_4 = x_5 + x_{11} + x_{13}$ | 18 | $y_0 = x_0 + y_6 + t_{16}$ |
| 6 | $y_2 = x_2 + x_6 + t_4$ | 19 | $y_{12} = x_6 + x_{12} + t_{16}$ |
| 7 | $t_6 = x_3 + x_6 + x_{14}$ | 20 | $t_{19} = x_4 + x_{12}$ |
| 8 | $y_{10} = x_0 + y_2 + t_6$ | 21 | $y_8 = x_2 + y_0 + t_{19}$ |
| 9 | $y_3 = x_7 + x_8 + t_6$ | 22 | $y_4 = x_0 + t_4 + t_{19}$ |
| 10 | $y_{11} = x_1 + t_0 + y_3$ | 23 | $t_{22} = x_{11} + y_{11}$ |
| 11 | $t_{10} = x_3 + x_7 + y_2$ | 24 | $y_{15} = x_5 + y_3 + t_{22}$ |
| 12 | $y_{13} = x_9 + x_{11} + t_{10}$ | 25 | $y_5 = x_3 + x_{15} + t_{22}$ |
| 13 | $y_{14} = t_2 + y_{10} + t_{10}$ | | |

As shown in Table 3, our proposed method leads to better results than the best-known XOR GCs and GEs given in the literature. In the case of involutory $4 \times 4$ MDS matrices over $\mathbf{F}_{2^4}$ with depth 5, the matrix $M_4$ presented in Example 15 offers 1 XOR operations improvement from the best-known result given in *Sarkar & Syed (2016)*. On the other hand, the lightest $4 \times 4$ involutory MDS matrix $M_3$ over $\mathbf{F}_{2^3}$ with the lowest known XOR count 13 (the lowest XOR count known so far) with depth 5 is presented in Table 3. In non-involutory MDS matrices section in Table 3, we compare the matrix $M_5$ presented in Example 16 with the best known XOR count results in the literature. The matrix $M_5$ as depth 3 is the lightest XOR count result with 19 XOR operations offering 1 XOR operations improvement.

**Table 7** The global optimization result of $M_6$ with 42 XORs and depth 4, where $x_i$s $[(x_0, x_1, \ldots, x_{31})]$, $y_i$s $[(y_0, y_1, \ldots, y_{31})]$ and $t_j$s represent input bits, output bits, and temporary variables, respectively.

| No. | Operation | No. | Operation |
|---|---|---|---|
| 1 | $y_8 = x_1 + x_8 + x_{23} + x_{25}$ | 23 | $y_4 = y_{14} + t_5 + t_{18}$ |
| 2 | $y_{24} = x_6 + x_8 + x_{17} + x_{24}$ | 24 | $y_{20} = x_{14} + x_{20} + t_{15} + t_{18}$ |
| 3 | $y_{14} = x_7 + x_{14} + x_{21} + x_{31}$ | 25 | $t_{24} = x_{15} + x_{19} + x_{26} + x_{29}$ |
| 4 | $y_{31} = x_5 + x_{15} + x_{16} + x_{31}$ | 26 | $y_{12} = y_{16} + y_{31} + t_{24}$ |
| 5 | $y_{15} = x_0 + x_{15} + x_{22} + x_{24}$ | 27 | $y_{26} = x_{16} + y_0 + y_{24} + t_{24}$ |
| 6 | $t_5 = x_7 + x_9 + x_{16} + x_{28}$ | 28 | $y_2 = x_2 + x_{26} + t_{12} + t_{24}$ |
| 7 | $y_{21} = y_{14} + y_{31} + t_5$ | 29 | $t_{28} = x_3 + x_{10} + x_{13} + x_{24}$ |
| 8 | $y_{25} = x_{25} + x_{18} + x_{28} + t_5$ | 30 | $y_{29} = x_{15} + y_{22} + t_{28}$ |
| 9 | $y_7 = x_{14} + t_5$ | 31 | $y_{10} = x_0 + y_8 + y_{17} + t_{28}$ |
| 10 | $t_9 = x_{10} + x_{15} + x_{24} + x_{29}$ | 32 | $y_{19} = x_{12} + x_{31} + t_{24} + t_{28}$ |
| 11 | $y_{22} = x_6 + x_{22} + t_9$ | 33 | $t_{32} = x_7 + x_{28} + t_{24} + y_2$ |
| 12 | $y_0 = x_0 + x_8 + x_{17} + t_9$ | 34 | $y_9 = t_5 + y_{16} + t_{32}$ |
| 13 | $t_{12} = x_{12} + x_{16} + x_{26} + x_{31}$ | 35 | $y_{28} = x_{21} + x_{31} + t_{32}$ |
| 14 | $y_5 = x_5 + x_{16} + x_{22} + t_{12}$ | 36 | $y_{18} = x_{14} + x_{25} + y_{25} + t_{32}$ |
| 15 | $y_{16} = x_0 + x_{24} + t_{12}$ | 37 | $t_{36} = x_8 + x_{20} + x_{27} + x_{30}$ |
| 16 | $t_{15} = x_8 + x_{13} + x_{27}$ | 38 | $y_{27} = x_6 + y_8 + y_{23} + t_{36}$ |
| 17 | $y_6 = x_6 + x_{23} + t_{15}$ | 39 | $y_{13} = y_6 + t_{36}$ |
| 18 | $y_{17} = x_1 + x_{17} + x_{25} + t_{15}$ | 40 | $y_3 = x_{15} + x_{29} + t_{28} + t_{36}$ |
| 19 | $t_{18} = x_4 + x_{11} + x_{25} + x_{30}$ | 41 | $t_{40} = x_{23} + x_{30} + y_{25} + t_{18}$ |
| 20 | $y_{23} = x_4 + x_7 + x_{23} + t_{18}$ | 42 | $y_{11} = t_5 + t_{40}$ |
| 21 | $y_{30} = x_{14} + x_{30} + t_{18} + y_{23}$ | | |
| 22 | $y_1 = x_8 + y_8 + y_{25} + y_{23}$ | | |

**Example 17.** *Let $\mathbf{F}_{2^8}$ be generated by the primitive element $\alpha$ which is a root of the primitive polynomial $x^8 + x^5 + x^3 + x^2 + 1$ (0x12d). Consider the $4 \times 4$ involutory MDS matrix $M_6$ as given below:*

$$M_6 = \begin{bmatrix} 1 & \alpha^{36} & \alpha^{254} & \alpha^{38} \\ \alpha^{254} & 1 & \alpha & \alpha^{254} \\ 1 & \alpha^{39} & 1 & \alpha^{36} \\ \alpha^2 & 1 & \alpha^{254} & 1 \end{bmatrix}$$

*over $\mathbf{F}_{2^8}/0x12d$. In fact, the involutory MDS matrix $M_6$ belongs to a class of which representative involutory MDS matrix is as follows:*

$$RIM_6 = \begin{bmatrix} 1 & \alpha^{145} & \alpha^{127} & \alpha^{20} \\ \alpha^{145} & 1 & \alpha^{20} & \alpha^{127} \\ \alpha^{127} & \alpha^{20} & 1 & \alpha^{145} \\ \alpha^{20} & \alpha^{127} & \alpha^{145} & 1 \end{bmatrix}$$

*The involutory MDS matrix $M_6$ with d-XOR count 162 $(= 66 + 4 \cdot 3 \cdot 8)$ can easily be obtained by applying the parameters $b_1 = \alpha^{146}$, $b_2 = \alpha^{127}$ and $b_3 = \alpha^{18}$ (and their inverses)*

*to representative involutory MDS matrix $RIM_6$. After applying BDKCI heuristic to the matrix $M_6$, we find the circuit for the matrix $M_6$ with 52 XOR operations and depth 5 using XOR2 and XOR3 gates. Moreover, the same matrix $M_6$ requires only 42 XOR operations with depth 4 using XOR2, XOR3 and XOR4 gates (please see Table 7 for details).*

In Table 4, we compare the matrix $M_6$ with the circuit implementations of ciphers given in the literature. When comparing the matrix $M_6$ with the previous best-known implementations of linear layers of some block ciphers, the involutory MDS matrix $M_6$ improves the results in GCs and GEs with the minimum circuit depths.

Note that in Tables 3 and 4 we do not only consider the construction of circuits with minimum GCs (*i.e.*, XOR gate counts) and GEs but also take into account the minimum circuit depth for low latency criterion. Since we stopped the algorithm after four hours of runtime for each matrix, lighter circuits can be found because the proposed new hybrid method allows us to generate all $4 \times 4$ involutory MDS matrices over $\mathbf{F}_{2^m}$. For further improvements, more runtime may be required.

## CONCLUSION AND FUTURE WORKS

In this article, we proposed a new hybrid method to generate all $4 \times 4$ involutory MDS matrices over $\mathbf{F}_{2^m}$. The proposed method reduces the search space complexity at the level of $\sqrt{n}$ by searching and finding representative involutory MDS matrices, where $n$ represents the number of all invertible $4 \times 4$ matrices over $\mathbf{F}_{2^m}$. In this respect, we were able to generate all $4 \times 4$ involutory MDS matrices over $\mathbf{F}_{2^3}$ and $\mathbf{F}_{2^4}$. For the finite field $\mathbf{F}_{2^8}$, the search space to generate all representative involutory MDS matrices is approximately $2^{64}$, which is still too high to be searched for. Nevertheless, one can easily generate $4 \times 4$ involutory and MDS matrices over $\mathbf{F}_{2^8}$ by focusing on representative involutory MDS matrices. In the future, if the number of parameters in $R_1$ presented in the "Proposed Method" section is reduced by eliminating some parameters and a new search form is obtained, then the search space for finding representative involutory MDS matrices for larger finite fields can be reduced. However, this new form is highly possible to be a more complex structure than $R_1$. As a result, we believe that our proposed method is not only useful for generating all involutory MDS matrices over $\mathbf{F}_{2^m}$, but it is also useful for other methods in the literature like the methods used for constructing lightweight involutory MDS matrices over the general linear groups GL $(m,\mathbf{F}_2)$.

## DECLARATIONS

All authors have read and agreed to the submitted version of the manuscript. Funding is not applicable. The authors declare no conflict of interest. Data sharing is not applicable to this article as no data sets were generated or analyzed during the current study.

## APPENDIX

Let $\mathbf{F}_{2^4}$ be generated by the primitive element $\alpha$ which is a root of the primitive polynomial $x^4 + x + 1$ (0x13). Consider $4 \times 4$ representative involutory MDS matrix as given below:

$$RIM_7 = \begin{bmatrix} 1 & \alpha^4 & \alpha^{11} & \alpha^{13} \\ \alpha^{11} & \alpha^{12} & \alpha^{11} & \alpha^{11} \\ \alpha^3 & \alpha^8 & \alpha^{12} & \alpha^4 \\ \alpha^5 & \alpha^3 & \alpha^{11} & 1 \end{bmatrix}$$

over $\mathbf{F}_{2^4}/0x13$. Then, one can generate $4 \times 4$ involutory and MDS matrix $M_7$ by applying the parameters $b_1 = \alpha^{11}$, $b_2 = \alpha^4$ and $b_3 = 1$ (and their inverses) to $RIM_7$ as follows:

$$M_7 = \begin{bmatrix} 1 & 1 & 1 & \alpha^{13} \\ 1 & \alpha^{12} & \alpha^4 & 1 \\ \alpha^{14} & 1 & \alpha^{12} & 1 \\ \alpha^5 & \alpha^{14} & 1 & 1 \end{bmatrix}.$$

The number of 1s in the matrix $M_7$ is 9, which is the maximum number of occurrences of 1s in $4 \times 4$ MDS matrices.

### Funding
The authors received no funding for this work.

### Competing Interests
Sedat Akleylek is the Section Editor of Cryptography, Security and Privacy.

### Author Contributions
- Gökhan Tuncay conceived and designed the experiments, performed the experiments, analyzed the data, performed the computation work, authored or reviewed drafts of the article, and approved the final draft.
- Fatma Büyüksaraçoğlu Sakallı conceived and designed the experiments, performed the experiments, analyzed the data, prepared figures and/or tables, authored or reviewed drafts of the article, and approved the final draft.
- Meltem Kurt Pehlivanoğlu conceived and designed the experiments, performed the experiments, analyzed the data, performed the computation work, prepared figures and/or tables, authored or reviewed drafts of the article, and approved the final draft.
- Gülsüm Gözde Yılmazgüç conceived and designed the experiments, performed the experiments, analyzed the data, authored or reviewed drafts of the article, and approved the final draft.
- Sedat Akleylek conceived and designed the experiments, performed the experiments, analyzed the data, authored or reviewed drafts of the article, and approved the final draft.
- Muharrem Tolga Sakallı conceived and designed the experiments, performed the experiments, analyzed the data, authored or reviewed drafts of the article, and approved the final draft.

### Data Availability
The source codes are available at GitHub and Zenodo:

\- https://github.com/mkurtpehlivanoglu/Hybrid_Method.

\- Meltem Kurt Pehlivanoglu, Vishnu Asutosh Dasu, & anubhab001. (2023). mkurtpehlivanoglu/Hybrid_Method: Initial Release (0.1.0). Zenodo. https://doi.org/10.5281/zenodo.8187919.

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
