# Peer review of "A new hybrid method combining search and direct based construction ideas to generate all 4 × 4 involutory maximum distance separable (MDS) matrices over binary field extensions"

_PeerJ Computer Science, doi:10.7717/peerj-cs.1577_

## Round 0.1 · original submission · Minor Revisions

Authors are asked to update the methodology with visualization and correct the minor grammar errors.

Reviewer 1 ·

Basic reporting

The content and language are in good condition. The references are cited well.

Experimental design

The research question is well-defined and the investigation has been performed to a high standard.

Validity of the findings

I have not checked all results but the ones I have checked are correct and I believe that the rest is also correct.

·

Basic reporting

The article has interesting results. This paper proposes a hybrid method combining search and direct construction of involutory MDS matrices in a finite field. The authors also conduct experiments to find these matrices, in addition, they also evaluate the number of XOR operations of the found matrices and compare with other works.
However, the authors need to correct the following comments.
- In the Introduction, the authors should introduce some other methods to build MDS matrix such as: circulant matrix, circulant-like matrix, and recursive MDS matrix.
- Lines 175-176, the authors should change to “The following are properties of an MDS matrix.”
- Line 187, should edit “a_i s” => “a_i's”.
- Line 198, edit “b_i s” => “b_i's”.
- With the representative involutory MDS matrix size 2×2, the authors proved in Lemma 3, and showed that the matrix RIM_(2×2) is the representative involutory MDS matrix of size 2×2. With size 4×4, the authors give the matrix R_1 as an involutory MDS matrix of size 4×4 and prove it through Lemma 6, however, it is necessary to put the matrix R_1 in front of Lemma 6.

Experimental design

Good

Validity of the findings

Good

·

Basic reporting

This paper contains a too much mathematical content without background, making it challenging for general readers to comprehend. In my opinion, the authors should consider incorporating additional explanations of the mathematics involved. There are several notations that are unclear, and the abstract itself is quite complex to grasp. Furthermore, the paper lacks background information or context for the equations presented. Here are some suggestions to improve the clarity and readability of the paper written in LaTeX:

Provide a clear and concise overview: Begin the paper by providing a brief overview of the problem statement and the main contribution of the proposed hybrid method for generating involutory Maximum Distance Separable (MDS) matrices.

Explain the significance of the research: Clearly explain why generating involutory MDS matrices over specific finite fields (such as $\mathbf{F}{2^3}$, $\mathbf{F}{2^4}$, and $\mathbf{F}_{2^8}$) is important and how it contributes to the field of cryptography or other relevant applications.

Define technical terms: Ensure that all technical terms, such as "involutory," "MDS matrices," and "search space complexity," are defined and explained in a concise manner. This will make the paper more accessible to readers who may not be familiar with the specific terminology.

Provide more details on the proposed hybrid method: Elaborate on the combination of search-based methods and direct construction methods in the proposed hybrid method. Explain how this approach reduces the search space complexity at the level of $\sqrt{n}$, where $n$ represents the number of invertible matrices to be searched for.

Present the results in a clear manner: Clearly present the results achieved by the proposed method, particularly in terms of generating involutory/non-involutory MDS matrices over $\mathbf{F}{2^3}$, $\mathbf{F}{2^4}$, and $\mathbf{F}_{2^8}$. Specify the XOR count and depth required for each matrix, highlighting their significance and efficiency.

Highlight the novelty of the findings: Emphasize the novelty of generating the lightest involutory/non-involutory MDS matrices known over the specified finite fields. Discuss how the proposed global optimization technique and the use of higher input XOR gates contribute to achieving these results.

Clarify the contribution related to Hadamard matrix: Clearly explain the new property of the Hadamard matrix and its role in generating a small subset of involutory MDS matrices. Highlight the specific value of this contribution, particularly regarding matrices with dimensions $2^k \times 2^k$, where $k>1$.

Proofread and revise for clarity: Ensure that the paper is thoroughly proofread for grammar, punctuation, and sentence structure errors. Simplify complex sentences and rephrase unclear or ambiguous statements to improve overall clarity and readability.

By implementing these suggestions, the paper will become more accessible, effectively communicate the proposed method and findings, and contribute to the existing body of knowledge in the field.

Experimental design

no experiment performed.

Validity of the findings

hard to find.

---

## Round 0.2 · accepted · Accept

The author has incorporated all comments and it is fine-tuned now.

Reviewer 1 ·

Basic reporting

I think the paper is now ready for publication

Experimental design

The research quality and the presentation style is very suitable for the journal

Validity of the findings

All the results seem valid and noval

·

Basic reporting

good

Experimental design

ok

Validity of the findings

good

Additional comments

good

·

Basic reporting

Authors updated the paper and no further comment from my side.

Experimental design

Good

Validity of the findings

Good